# Sputum-Rheology-Based Strategy for Guiding Azithromycin Prescription in COPD Patients with Frequent Exacerbations: A Randomized, Controlled Study (“COPD CARhE”)

**DOI:** 10.3390/biomedicines11030740

**Published:** 2023-03-01

**Authors:** Jeremy Charriot, Maeva Zysman, Laurent Guilleminault, Mathilde Volpato, Aurelie Fort-Petit, Isabelle Vachier, Jeremy Patarin, Carey Suehs, Engi Ahmed, Nicolas Molinari, Arnaud Bourdin

**Affiliations:** 1Faculty of Medicine, University of Montpellier, PhyMedExp INSERM U1046, 34090 Montpellier, France; 2CHU Montpellier, Respiratory Diseases Department, Arnaud de Villeneuve Hospital, 34090 Montpellier, France; 3Centre de Recherche Cardio-Thoracique de Bordeaux, Univ-Bordeaux, INSERM U1045, 33000 Pessac, France; 4Department of Respiratory Medicine, Faculty of Medicine, Toulouse University Hospital, 31300 Toulouse, France; 5Medecine Biologie Meditérrannée, 34000 Montpellier, France; 6Rheonova, Domaine Universitaire, 38400 Saint Martin d’Hères, France; 7Department of Medical Information, Montpellier University Hospitals, La Colombière Hospital, 34090 Montpellier, France

**Keywords:** COPD, exacerbations, sputum rheology, azithromycin, mucus, eosinophil

## Abstract

(1) Background: We have previously shown that sputum rheology can discriminate between patients with COPD and other muco-obstructive lung diseases, and that it is correlated with mucin content and sputum eosinophilia. We now hypothesize that it could be a more-accurate guide than clinical evaluation for the prescription of azithromycin to prevent exacerbations of COPD and to reduce exposure to antibiotics; (2) Methods: “COPD CaRhe” is a multicentric, randomized, controlled trial comparing outcomes in two parallel arms (36 vs. 36 patients). Patients will be recruited in the university hospitals of Montpellier, Bordeaux, and Toulouse, in France, and they should have a diagnosis of COPD with frequent exacerbations (≥3/year). Enrollment will occur during a routine visit to a respiratory department, and follow-up visits will occur every 3 months for a period of 1 year. At each visit, a 3-month prescription of azithromycin will be provided to those patients who obtain a score of <70 on the Cough and Sputum Assessment Questionnaire (CASA-Q) or a critical stress score of σc > 39 on a rheological assessment of sputum, depending upon their randomization group. The primary outcome will be the number of exacerbations of COPD; (3) Discussion: By using sputum rheology, the COPD CaRhe study may provide clinicians with an objective biomarker to guide the prescription of azithromycin while reducing the cumulative exposure to macrolides.

## 1. Introduction

Chronic obstructive pulmonary disease (COPD) is a devastating and currently irreversible chronic disease that affects approximately 10–13% of the European population [1,2]. Its prevalence and incidence are increasing [3]. Chronic bronchitis (chronic cough and sputum) is a feature of COPD which is predictive of acute exacerbations of COPD (AECOPD) [4]. These events are part of the natural history of this disease and a marker of poor prognosis [5]. For instance, in France, more than 8% of patients hospitalized with acute AECOPD die during their stay [6]. Currently, therapeutic strategies aim to decrease their annual rate.

For a long time [7], several elements have been analyzed in sputum to determine the stable or exacerbated character of COPD, including its volume, its color, and its purulence. Sputum is classically attributed to excessive mucus production in the respiratory tract. This hyperproduction of mucus is now even thought to play a major role in distal airway (bronchiolar) obstruction, which is then complicated by emphysema [8]. While mucus production is a classic symptom of exacerbations, this symptom persists in the chronic form in slightly more than half of patients, with a rather negative significance, even if this remains controversial [4,9]. (Chronic bronchitis is defined by the presence of cough and sputum for 3 consecutive months over a period of at least 2 years.) This excess of mucus is responsible for chronic sputum that constitutes a considerable dimension of the burden carried by patients in this disease. However, at the moment, in France, there is no treatment reimbursed by health insurance that specifically targets this symptom, mostly due to lack of evidence or efficacy.

Multiple components vary in this sputum, and they are independent of the intrinsic quality and quantity of mucus [10] produced by the epithelium: microorganisms, epithelial cellular debris, polynuclear DNA, and other inflammatory cells. These changes are likely to modulate the rheology of the mucus [11]. Mucus is a complex, viscoelastic component consisting essentially of water, ions, and proteins, some of which (with the mucins MUC5AC and MUC5B being the most commonly represented) can modulate the biological and biophysical properties of mucus. Goblet cell hyperplasia and metaplasia would thus be the primum movens of chronic bronchitis [12,13]. This airway remodeling is partially explained by an increased production of certain cytokines (IL-4, IL-13, etc.) and mediators of epithelial or systemic origin [14]. The complexity of the pathophysiology of COPD is difficult to portray; nevertheless, it may be approached through the elaboration of biological network models that focus on cell proliferation, cell fate, or inflammation [15]. All of these processes are implied in the hyperproduction of mucus. Interestingly, azithromycin has been shown to decrease neutrophil chemotaxis in the lower respiratory tract [16], to improve the phagocytic capacity of alveolar macrophages [17], and to reduce some markers of systemic inflammation. However, equally convincingly, azithromycin has been shown to inhibit the hypersecretion of mucus by airway epithelial cells [18], which may explain why this macrolide engenders fewer exacerbations. Indeed, in clinical trials, the use of long-term azithromycin at a reduced-dosage regimen (either 250 mg/d or 500 mg/3 times per week) significantly reduced the annual rate of AECOPD, respectively, by 27% and 42% in two large, randomized, double-blind, controlled trials against placebo [19,20]. Moreover, it also proved efficient in improving the cough-specific health status of patients with COPD [21] and the Leicester Cough Questionnaire (LCQ) scores of patients with resistant cough [22]. As a result, azithromycin is now proposed in international and French guidelines as a therapeutic tool for the prevention of exacerbations, in addition to optimal standard management [23,24]. However, the pathophysiological link between azithromycin administration and the reduction of COPD exacerbations has not been established. As macrolides may cause infrequent, but significant, adverse effects (e.g., deafness; cardiac conduction disorders) and may increase microbial resistance, prescriptions of this type of molecule must be properly targeted. Nevertheless, the COPD patient population receiving azithromycin (or macrolides in general) is poorly defined and far from specifically “targeted”.

The objective selection of a subpopulation of COPD patients, based on altered or unaltered sputum rheology (now easily quantifiable within minutes), would provide a more effective treatment algorithm than the current standard management. In patients with COPD, asthma, and non-CF bronchiectasis, our team recently demonstrated that sputum rheology (elastic modulus G’, viscous modulus G’’, and critical stress σc), was correlated with MUC5AC and MUC5B mucin content and sputum eosinophilia [25]. We have also shown that sputum rheology can predict sputum eosinophilia at a rate superior or equal to 1.25%, a value that correlated with exacerbations requiring corticosteroids and a reduction in FEV1 in the SPIROMICS cohort [26].

Here, we hypothesize that sputum rheology could guide the prescription of azithromycin in a more appropriate manner than does clinical evaluation, so as to reduce both AECOPD and exposure to antibiotics.

## 2. Materials and Methods

### 2.1. Objectives

#### 2.1.1. Primary Objective

The primary objective of this trial is to compare the exacerbation rates over 12 months between the comparator arm, i.e., a group of COPD patients with frequent exacerbations treated according to clinical evaluation using the Cough and Sputum Questionnaire (CASA-Q; see below for details), and the experimental arm, comprising a similar group in which azithromycin is prescribed according to sputum rheology.

#### 2.1.2. Secondary Objectives

To compare the two arms, we will study:-the exacerbation rates by severity (as observed throughout the study);-the evolution of symptoms, sputum rheology, and lung function (via repeated measurements collected every 3 months);-the medication usage and adverse events (follow-ups for the duration of the study);-the patient trajectories during follow-ups;-the overall clinical improvement achieved by the end of the study and changes in quality of life (via repeated measurements every 3 months); and-the changes in biomarkers of interest (baseline versus end of study).-We will compare the exacerbation rates, throughout the study, of the experimental arm without azithromycin prescription, and of the comparator arm without azithromycin; and between the experimental arm with azithromycin prescription and the comparator arm with azithromycin.-We will evaluate patients in the experimental arm who would have been managed differently using the CASA-Q versus rheology.

### 2.2. Study Design

COPD CaRhe is a parallel-group randomized trial (36 vs. 36 patients with COPD and frequent exacerbations). It will compare a group of patients treated according to clinical evaluation (and azithromycin will be prescribed according to the Cough and Sputum Assessment Questionnaire (CASA-Q), if patients score < 70) versus a similar group of patients in which azithromycin prescription will be guided by sputum rheology (if the σc > 39). The chosen cut-offs of 70 for the CASA-Q and 39 for the σc correspond to the expected medians of these variables in our population of interest. (See Figure 1).

The CASA-Q is a questionnaire validated in French which aims to rate cough and sputum and their respective impacts on quality of life. According to the national [24] and international [23] recommendations in force, the prescription of azithromycin is conditioned upon the specialist’s free, subjective appreciation of respiratory symptoms, including cough and sputum, in patients with COPD and frequent exacerbations. These symptoms will be assessed here by the CASA-Q [27,28,29,30].

For the variable “critical stress σc”, the threshold of 39 was determined based on a previously published ancillary study [25].

### 2.3. Setting and Participant Eligibility

All patients consecutively attending the outpatient clinic in the Respiratory Diseases Departments of the Montpellier, Bordeaux, and Toulouse University Hospitals (France) who meet the eligibility criteria will be prospectively enrolled. Adults aged 40–85, with a confirmed diagnosis of COPD according to international guidelines, are eligible. COPD patients must have a “frequent exacerbation” phenotype, i.e., at least three exacerbations over the past 12 months before inclusion, despite optimal treatment according to the Global Initiative for Chronic Obstructive Lung Disease [23] guidelines. All inclusion and exclusion criteria are summarized in Table 1.

Basic demographics (age, sex, weight, height, and body mass index) and comorbidities (specifically, the starting dates for renal insufficiency, diabetes, diffuse interstitial pneumonia, or ischemic heart disease), if present, will be used to describe the population at baseline. A listing of any concomitant treatments will be maintained for each patient throughout the study.

### 2.4. Interventions

In both the comparator arm and the experimental arm, patients will receive optimal background treatment for COPD according to current national and international guidelines.

#### 2.4.1. Comparator Arm

Patients randomized to the comparator arm of the study will receive azithromycin in case of severe sputum symptoms, here defined by a CASA-Q sputum symptom score of <70, so as to homogenize practices between centers. The CASA-Q will be scored every 3 months.

-If the patient presents a sputum symptom score of <70, azithromycin will be initiated for 3 months. This prescription will be renewable every 3 months during the 12 months of follow-up planned in this study if the patient continues to obtain a sputum symptom score of <70.-If the patient has a sputum symptom score of >70, management will not be changed.

#### 2.4.2. Experimental Arm

Patients randomized to the experimental arm of the study will receive azithromycin according to sputum rheology. Sputum rheology will be quantified every 3 months using a Rheomuco device.

-If sputum has a critical stress σc >39, azithromycin will be initiated for 3 months. This prescription will be renewable every 3 months during the 12 months of follow-up planned in this study.-In the absence of sputum, azithromycin will not be prescribed.

### 2.5. Randomization and Blinding

To optimize the comparability between the two groups, a blocked randomization with randomly selected block sizes will be used to assign patients to treatment groups using a 1:1 allocation ratio. Due to the sample size, stratification will only be performed according to each center. The other variables of interest (e.g., the number of exacerbations in the 12 months prior to inclusion, gender, background treatment, etc.) will serve as adjustment variables in the statistical analysis.

Given the scope of this first study in the field, a level of blinding involving a placebo would not be appropriate. Nevertheless, by blinding the sputum rheology and CASA-Q results for the duration of the study, the blinding allowed by randomization will be ensured. A “guess the arm” question will be asked of investigators and patients at the end of the study to assess the level of blinding.

Data required at the baseline visit, namely sputum rheology and CASA-Q responses, will be promptly entered into the eCRF by a designated person at each center, as these data will condition the allocation of the treatment.

A web-based application within the eCRF (Ennov Clinical) will be created for the purposes of the study and will ensure the randomization of patients. The creation of the eCRF, the sequence, and the allocation modules will be supervised by the Department of Clinical Research and Epidemiology of Montpellier University Hospital. In addition to randomization, the eCRF will also be parameterized to automatically apply the treatment algorithm specific to each study arm. The randomization of patients will be performed by the participating investigators through specific modules in the eCRF. The randomization will be performed after the data entry corresponding to the sputum rheology or the CASA-Q.

### 2.6. Predicted and Explanatory Variables

Patient-specific measures and time frames are summarized in Table 2.

### 2.7. Sputum Collection

A spontaneous or induced, unselected sputum sample will be collected for each individual patient. All samples will be rigorously processed following the same procedure, as previously described. Within 2 h of collection, rheology parameters will be measured, and then cytology assessments will be performed (see the following paragraph). In parallel, a part of the sample will be sent to routine, direct bacterial examination and culture.

### 2.8. Sputum Rheology

Rheology will be assessed on 720 mL of 1 fresh sputum sample, as previously described by Patarin et al. [31] Briefly, after gentle vortexing at low velocity for 1 min, rheometric analysis will be performed using the Rheomuco^®^ rheometer (Rheonova, Grenoble, France) at 37 °C, with an oscillatory frequency of 1 Hz and an increasing strain. Measurements of the critical stress threshold (σc) will be obtained as needed to determine the prescription of azithromycin. Other parameters will also be collected (such as elastic modulus (G’; Pa), viscous modulus (G’’; Pa), etc.).

### 2.9. Pulmonary Function Tests

Spirometry and plethysmography will be carried out as recommended by the European Respiratory Society using BodyBox equipment (MediSoft BodyBox 5500, Sorinnes, Belgium). With the exception of the FEV1/FVC ratio, which is calculated as liters/liters, all outcomes (forced expiratory volume in 1 s (FEV1), forced vital capacity (FVC), total lung capacity (TLC), and residual volume (RV)) will be calculated as both % predicted values and as liters.

### 2.10. Blood Tests

A blood cell count, and especially a blood eosinophil count, will be performed at baseline and 12 months, as routinely performed in each university hospital. Identically, an assessment of the serum club cell secretory protein (CCSP; ng/mL) level will be performed, as determined by an enzyme-linked immunosorbent assay, according to the manufacturer’s instructions (Biovendor, Brno, Czech Republic). A liver blood test will be performed at baseline to rule out severe hepatic insufficiency.

### 2.11. Quality-of-Life Questionnaires

The following questionnaires will be administered to each patient at each visit during the follow-up: the SF-36, the EQ-5D-5L, and the St. George’s Respiratory Questionnaire.

### 2.12. Prescription of Azithromycin

The prescription of azithromycin will be standardized in both arms: 500 mg (250 mg x 2/d) 3 days per week for 3 months, with possible renewals during the 12 months of follow-up. Azithromycin tablets may be taken with or without meals.

As the information contained in the marketing authorizations is likely to change, it is advisable that clinicians make sure at the time of prescription that the contraindications, warnings and precautions for use, and drug interactions are respected by referring to the information available on the public drug database, accessible on the Internet at the following address: http://base-donnees-publique.medicaments.gouv.fr/ (accessed on 22 February 2008).

### 2.13. Ethics

This study was approved by the Comité de Protection des Personnes IDF X—Ile de France X (N° EudraCT 2019-004182-41) and approved by the ANSM (Agence National de Sécurité du medicament et des produits de Santé) on 29 November 2021, and it has been registered on Clinicaltrials.gov (NCT04339270). In accordance with French regulations for observational studies, verbal and written consent will be required, and patients can participate as long as they are correctly informed and do not oppose the data analysis.

### 2.14. Sample Size and Statistics

In the comparator arm, we expect an exacerbation rate of 3.59/year [32]. The choice of standard deviation is most conservative in GOLD classes C–D, i.e., SD = 2.5 [33]. To detect a 50% reduction (the reduction of interest for this first-of-its-kind study) in the exacerbation rate in the experimental arm, with a two-sided, first-species risk of 0.05 and 1-beta power = 0.80, a total of 62 patients is required (31 per group). Taking into account a 10% loss of data (due to screening failures and study dropouts), we propose the inclusion of a total of 72 patients in this study (36 in each arm).

Descriptive statistics will be provided as medians, followed by their interquartile ranges for quantitative variables, and as numbers (%) for qualitative variables. Pairwise comparisons between groups will be performed using Conover’s tests or Fisher’s exact tests, corrected for false discovery rates (FDRs). Multivariable, stepwise regression will then be performed to explain potential differences between groups.

### 2.15. Study Schedule

-Duration of the regulatory process: 6–9 months;-Duration of the inclusion period: 24 months;-Duration of the participation of each participant: 12 months;-Duration of the statistical analysis and writing of the final report: 12 months;-Anticipated duration of the study (inclusion and follow-up): 36 months;-Anticipated total study duration: 48 months;-Number of visits from the beginning of the study: 7 per patient (see Table 2).

## 3. Discussion

COPD CaRhe will be the first clinical study to use sputum rheology to guide the prescription of a medication known to reduce the rate of exacerbation in COPD. This trial might lead to the implementation of sputum rheology in routine clinical practice. Thus, it could regularly assist the respiratory physician in making therapeutic choices and help in the reduction (or avoidance) of patients’ exposure to antibiotics. It will also give us insights into the evolution of patients’ sputum rheology over a long period of time. Moreover, we will have the opportunity to analyze multiple variables, such as serum eosinophils and CCSP, to explain the determinants of this rheology.

Cough and chronic sputum are symptoms that are still poorly understood and particularly poorly managed at present. Finely modulating this feature in a supervised manner therefore seems of utmost interest.

This study could have great implications for the clinical routine since it could introduce a new objective tool to assess the sputum of COPD patients; indeed, until now, sputum has been most often assessed on a daily basis by pulmonologists worldwide via simple questions (Do you cough? Do you produce phlegm?), sometimes by bacteriological examination (often negative), more rarely by cytology.

## Figures and Tables

**Figure 1 biomedicines-11-00740-f001:**
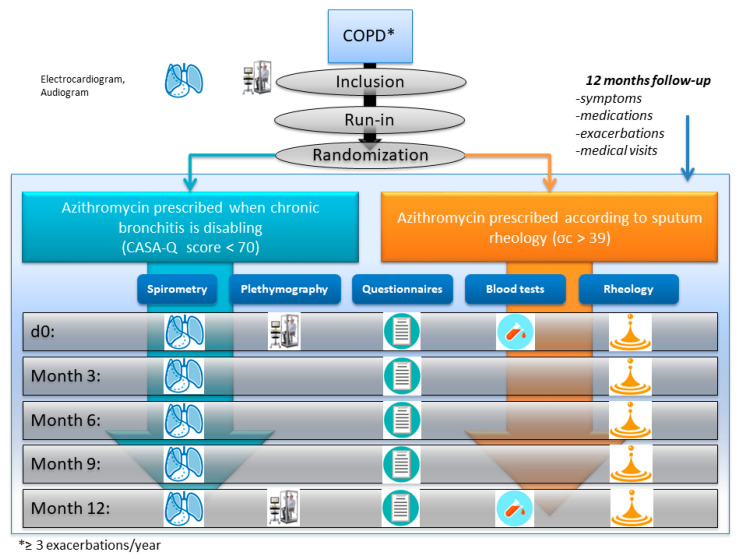
Study Design.

**Table 1 biomedicines-11-00740-t001:** Eligibility Criteria.

Inclusion Criteria	Exclusion Criteria
Subjects aged 40–85 years	Patients who are pregnant or breastfeeding
Diagnosis of COPD for at least 1 year	Patients who are prisoners or under other forms of judicial protection
Optimal treatment according to GOLD guidelines	Patients under any form of guardianship
≥3 exacerbations in the past 12 months	Participation in another interventional protocol
Spontaneous or induced sputum production	Patients who have received azithromycin in the past 3 months
Electrocardiogram: corrected QTC < 450 ms in men; QTC < 470 ms in women	Patients found to have bronchiectasis in a chest CT scan
Normal audiogram	Patients who have a known hypersensitivity to azithromycin, or any other macrolide
Written and signed informed consent form	Concomitant use of a medication contraindicated with azithromycin (dihydroergotamine, ergotamine, cisapride, or colchicine)
Subjects must be able to attend all planned visits and comply with all test procedures	Patients with other respiratory diseases or associated lung infections
Beneficiary of, or affiliated with, the French social security system	Severe hepatic insufficiency and severe cholestasis
	GFR < 40 mL/min
	Patients with hematological malignancies who have undergone allogeneic hematopoietic stem-cell transplantation
	Patients with galactose intolerance, Lapp lactase deficiency, or glucose-galactose malabsorption syndrome (a rare hereditary disease)

COPD: chronic obstructive pulmonary disease; GFR: glomerular filtration rate; GOLD: Global Initiative for Chronic Obstructive Lung Disease.

**Table 2 biomedicines-11-00740-t002:** Patient-specific measures and time frames.

	Pre-Inclusion	Inclusion	Run-In	Follow-Up				
	Vpre	Vinc *	RI	V0	V3	V6	V9	V12
	d–60 to d–28	d–30 to d–15	15 Days	d0	3 Mo ± 7 d	6 Mo ± 7 d	9 Mo ± 7 d	12 Mo ± 7 d
**Eligibility/Inclusion**								
Verification of Eligibility Criteria	✓	✓						
Oral and Written Information	✓							
Electrocardiogram		✓						
Liver Blood Test		✓						
Audiogram		✓						
Written Consent		✓						
Pregnancy Test (-hcg), if applicable		✓		✓	✓	✓	✓	✓
Verification of Patient Stability (exacerbations)			✓					
**Experimental Intervention**								
Randomization				✓				
Standard Management, or according to sputum rheology				✓				
**Assessments**								
Checking Patient’s Logbook: symptoms, exacerbations, unexpected use of drugs or medical resources		✓						
Plethysmography		✓						✓
Spirometry		✓		✓	✓	✓	✓	✓
Questionnaires				✓	✓	✓	✓	✓
Blood Test: blood cell count, CCSP *				✓				✓
Sputum Analysis: rheology, bacteriology		✓		✓	✓	✓	✓	✓
Side Effects			✓	✓				

* CCSP: Club cell secretory protein.

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
