# Peer review of "Sputum-Rheology-Based Strategy for Guiding Azithromycin Prescription in COPD Patients with Frequent Exacerbations: A Randomized, Controlled Study (“COPD CARhE”)"

_biomedicines, 2023, doi:10.3390/biomedicines11030740_

Round 1

Reviewer 1 Report

This paper describes the protocol for a clinical study aimed at assessing the optimum way to determine responders to azithromycin therapy for frequent exacerbations. The scientific rationale is fully described, and clear, and the methods for patient enrolment, assessment and statistical analysis appear appropriate. I have no major comments on the manuscript, and agree with the authors that the clinical problem is an important one, and that work like this may add to knowledge to solve it. It is an unusual paper style for this specific journal, in my experience of it to date.

Author Response

Dear Editor, dear reviewers, we would like to thank you very much for your time and your valuable comments.

Please find below our proposed responses to each of them (In bold and italic style):

Best regards

Jeremy Charriot for all the COPD CaRhe Team

  • Thank you very much for your remarks. We fully agree that this kind of paper may be unusual for this journal, that's why we made it clear to the editor what we were planning to write at the time of the invitation. We still are convinced our paper fits the aim of this special issue, as more and more papers are published nowadays about sputum rheology, especially in severe asthma and non-CF bronchictasis, with great papers from the famous team of Chapel Hill and the Marsico Lung Institute. We do think our translationnal approach, using rheology as a therapeutic strategy, is quite innovative in COPD and we look forward to sharing the results of this study soon with your readers.

Reviewer 2 Report

In this post study the authors hypothesize that sputum rheology could guide azithromycin prescription in a more appropriate manner than clinical evaluation in order to reduce AECOPD and exposure to antibiotics. They observed that by using sputum rheology, the COPD CaRhe study may give to clinicians an objective biomarker to guide the prescription of azithromycin while reducing the cumulative exposure to macrolides.

The manuscript is well organized. The experimental design has been well conceived and carried out. Methods are appropriate, results are clearly described and illustrated, as well as properly discussed. References are relevant and sufficiently updated. The quality of figures and English language is satisfactory. This paper can be useful for Biomedicines readers, because it provides very interesting information within the current context of only few published studies.

Author Response

Response to the reviewers

Dear Editor, dear reviewers, we would like to thank you very much for your time and your valuable comments.

Please find below our proposed responses to each of them (In bold and italic style):

Best regards

Jeremy Charriot for all the COPD CaRhe Team

  • We warmly thank you for your comment. We are convinced that this translational approach based on rheology will prove interesting for a very large audience, for basic scientists as well as for clinicians. We do think that rapid-on site rheology could be a useful “bedside tool” for clinical evaluation in muco-obstructive lung diseases.

Reviewer 3 Report

The article is deal with clinical trial protocol “Sputum rheology-based strategy to guide Azithromycin prescription in COPD patients with frequent exacerbations”. The topic discussed is very important for the treatment and prevention of complications of COPD.

I would like to make a few comments:

1. In the introduction, it is necessary to give a broader understanding of the problem and existing approaches to the diagnosis, therapy and prevention of complications. In particular, you may specify:

1) Modern approaches using biological network models are used, offering a holistic way to understand the biological processes associated with COPD in order to develop treatment recommendations based on basic research.

Quote the article, please:

sbv IMPROVER project team (in alphabetical order); Boue S, Fields B, Hoeng J, Park J, Peitsch MC, Schlage WK, et al. Enhancement of COPD biological networks using a web-based collaboration interface. F1000Res. 2015 Jan 29;4:32. doi: 10.12688/f1000research.5984.2. PMID: 25767696

2)    On the other hand, there is empirical evidence supporting the clinical efficacy of azithromycin in the prevention of complications and treatment of COPD.

Quote the articles, please:

 Berkhof FF, Doornewaard-ten Hertog NE, Uil SM, Kerstjens HA, van den Berg JW. Azithromycin and cough-specific health status in patients with chronic obstructive pulmonary disease and chronic cough: a randomised controlled trial. Respir Res. 2013 Nov 14;14(1):125. doi: 10.1186/1465-9921-14-125.

Hodgson D, Anderson J, Reynolds C, Oborne J, Meakin G, Bailey H, Shaw D, Mortimer K, Harrison T. The Effects of Azithromycin in Treatment-Resistant Cough: A Randomized, Double-Blind, Placebo-Controlled Trial. Chest. 2016 Apr;149(4):1052-60. doi: 10.1016/j.chest.2015.12.036.

2. Experiments on humans are not conducted, so instead of "experimental group" should be written "control group".

3. Inclusion criteria should take into account the status of a smoker (for example: do not smoke).

https://westessexccg.nhs.uk/your-health/medicines-optimisation-and-pharmacy/clinical-guidelines-and-prescribing-formularies/03-respiratory-system/3899-copd-prophylactic-use-of-azithromycin/file

4.    Before starting this treatment it is necessary to measure a heart trace (ECG), to check the rhythm of your heart, blood tests to check the liver function. In order to monitor whether azithromycin affects the liver blood tests to check the liver function should be done during treatment in control group.

https://www.uhcw.nhs.uk/download/clientfiles/files/Azithromycin%20and%20Chronic%20Obstructive%20Pulmonary%20Disease%20(COPD)%20290421_doc%20FINAL.pdf

5.    In the group receiving azithromycin, it is necessary to provide a sputum sample, to check for antibiotic resistant organisms before treatment.

6.    In the footnote of table 2 indicate, please: CCSP - Club Cell Secretory Protein. And blood test to check liver functions should be measured during treatment.

7.    In the item 2.8. Blood tests it is necessary to check the liver function during investigation in control group.

8.    In the item 2.13 Study Schedule it is necessary to indicate numbers and days of visits from the beginning of the treatment.

Author Response

Response to the reviewers

Dear Editor, dear reviewers, we would like to thank you very much for your time and your valuable comments.

Please find below our proposed responses to each of them (In bold and italic style):

Best regards

Jeremy Charriot for all the COPD CaRhe Team

  1. In the introduction, it is necessary to give a broader understanding of the problem and existing approaches to the diagnosis, therapy and prevention of complications. In particular, you may specify:

1) Modern approaches using biological network models are used, offering a holistic way to understand the biological processes associated with COPD in order to develop treatment recommendations based on basic research.

Quote the article, please:

sbv IMPROVER project team (in alphabetical order); Boue S, Fields B, Hoeng J, Park J, Peitsch MC, Schlage WK, et al. Enhancement of COPD biological networks using a web-based collaboration interface. F1000Res. 2015 Jan 29;4:32. doi: 10.12688/f1000research.5984.2. PMID: 25767696

=> Thank you for the reference which is highly relevant in our context. It has been added line 69.

2)    On the other hand, there is empirical evidence supporting the clinical efficacy of azithromycin in the prevention of complications and treatment of COPD.

Quote the articles, please:

 Berkhof FF, Doornewaard-ten Hertog NE, Uil SM, Kerstjens HA, van den Berg JW. Azithromycin and cough-specific health status in patients with chronic obstructive pulmonary disease and chronic cough: a randomised controlled trial. Respir Res. 2013 Nov 14;14(1):125. doi: 10.1186/1465-9921-14-125.

Hodgson D, Anderson J, Reynolds C, Oborne J, Meakin G, Bailey H, Shaw D, Mortimer K, Harrison T. The Effects of Azithromycin in Treatment-Resistant Cough: A Randomized, Double-Blind, Placebo-Controlled Trial. Chest. 2016 Apr;149(4):1052-60. doi: 10.1016/j.chest.2015.12.036.

=> Thank you, we have added those references line 81. They support well the rationale of our study

  1. Experiments on humans are not conducted, so instead of "experimental group" should be written "control group".
  2. Inclusion criteria should take into account the status of a smoker (for example: do notsmoke).

https://westessexccg.nhs.uk/your-health/medicines-optimisation-and-pharmacy/clinical-guidelines-and-prescribing-formularies/03-respiratory-system/3899-copd-prophylactic-use-of-azithromycin/file

 => Thank you for your remark. We did not chose this criterion for eligibility as some COPD patients have already succeeded in their attempt to stop smoking, but still have frequent exacerbations and so may still have an indication for Azithromycin. However, smoking status is an explanatory variable that is systematically recorded at the inclusion and at each visit in our electronic logpad. It will be analyzed as a potential factor influencing sputum rheology.

  1. Before starting this treatment it is necessary to measure a heart trace (ECG), to check the rhythm of your heart, blood tests to check the liver function. In order to monitor whether azithromycin affects the liver blood tests to check the liver function should be done during treatment in control group.

=> Your remark is relevant, and as mentioned in Table 2., which summarized the different interventions performed at each visit, an EKG is scheduled at baseline to rule out long QT syndrome which is an exclusion criterion (see Table 1.).

=> A liver blood test is performed at baseline to rule out severe hepatic insufficiency, which is indeed an exclusion criterion (see Table 1.). For more readability, we have detailed that line 252.

https://www.uhcw.nhs.uk/download/clientfiles/files/Azithromycin%20and%20Chronic%20Obstructive%20Pulmonary%20Disease%20(COPD)%20290421_doc%20FINAL.pdf

  1. In the group receiving azithromycin, it is necessary to provide a sputum sample, to check for antibiotic resistant organisms before treatment.

=> This is highly relevant and is planned in our design. We have added it in the table 2. (it was an omission). However, in our experience, we may have limited data concerning this as we need at least 720uL for sputum rheology, that can’t be used afterwards for microbiology (as contamination is too important), so we are exposed to have missing data for the microbiology variable.

  1. In the footnote of table 2 indicate, please: CCSP - Club Cell Secretory Protein. And blood test to check liver functions should be measured during treatment.

=> We have modified our table 2. accordingly to your remark. (see our response to your remark #4)

  1. In the item 2.8. Blood tests it is necessary to check the liver function during investigation in control group.

=> We have added a sentence concerning this issue.

  1. In the item 2.13 Study Schedule it is necessary to indicate numbers and days of visits from the beginning of the treatment.

=> We have added that in this section.
